



# Characterization of the cloud microphysical and optical properties and aerosol-cloud interaction in the Arctic from in situ ground-based measurements during the CLIMSLIP-NyA campaign, Svalbard

Gwennolé GUYOT[1], Frans OLOFSON[1], Peter TUNVED[2], Christophe GOURBEYRE[1], Guy FEVBRE[1], Régis DUPUY[1], Christophe BERNARD[1], Gérard ANCELLET[3], Kathy LAW[3], Boris QUENNEHEN[4], Alfons SCHWARZENBOECK[1], Kostas ELEFTHERIADIS[5], Olivier JOURDAN[1]

[1] Laboratoire de Météorologie Physique (LaMP), Université Blaise Pascal (UBP), OPGC, CNRS UMR 6016, Clermont-Ferrand, France
[2] Department of Applied Environmental Science (ITM), Stockholm University, Stockholm, Sweden
[3] Laboratoire Atmosphère, Milieux et Observations Spatiales (LATMOS), IPSL, UPMC, CNRS UMR 8190, Paris
[4] Laboratoire de Glaciologie et Géophysique de l'Environnement (LGGE), Université Grenoble Alpes/CNRS, 38041 Grenoble, France
[5] Environmental Radioactivity Laboratory, Institute of Nuclear and Radiological Science & Technology, Energy & Safety, Attiki, Greece

## *Abstract*

This study will focus on cloud microphysical and optical characterization of three different types of episodes encountered during the ground based CLIMSLIP-NyA campaign performed in Ny-Alesund, Svalbard: the Mixed Phase Cloud (MPC), snow precipitation and Blowing Snow (BS) events. These in situ cloud measurements will be combined with aerosol measurements and air mass backtrajectory simulations to qualify and parameterize the arctic aerosol cloud interaction and to assess the influence of anthropogenic pollution transported into the Arctic.

The results show a cloud bimodal distribution with the droplet mode at 10 µm and the crystal mode centered at 250 µm, for the MPC cases. The precipitation cases presents a crystal distribution centered around 350 µm with mostly of dendritic shape. The BS cases show a higher concentration but smaller crystals, centered between 150 and 200 µm, with mainly irregular crystals.

A "polluted" case, where aerosol properties are influenced by anthropogenic emission from Europe and East Asia, was compared to a "clean" case with local aerosol sources. These anthropogenic emissions seem to cause higher Black Carbon, aerosol and droplet concentrations, a more pronounced accumulation mode, smaller droplet sizes and a higher activation fraction $F_a$. Moreover, the activation diameter decreases as the droplet diameter increases and $F_a$ increases showing that smaller particles are activated and droplets grow when the aerosol number decreases. This is in agreement with the first (Twomey) and second (Albrecht) aerosol indirect effect. The quantification of the variations of droplet concentration and size leads to $IE$ (Indirect Effect) and $NE$ (Nucleation Efficiency) coefficients values around 0.2 and 0.43, respectively. These values are close to those found by other studies in the arctic



region which confirms these parameterizations of arctic aerosol-cloud interaction in climate
models.

## *1 Introduction*

The Arctic is a region where the surface warming is faster than the global average warming,
associated with, in particular, a rapid melting of the sea ice in summer (Vaughan et al., 2013).
This is the so-called arctic amplification. Several studies indicate that the arctic warming is
mainly of anthropogenic origin (e.g., Mc Guire et al., 2006, Serreze and Francis, 2006). The
arctic amplification is due to several positive feedbacks specific to the Arctic, the most
important being the sea ice melting feedbacks (Screen and Simmonds, 2010). Changes in
atmospheric and oceanic circulation, cloud properties (especially cloud cover) and atmospheric
water vapor amount are highly expected but their quantification remains uncertain. Specially,
the effects of clouds dominate the intermodal standard deviation of a temperature rise due to an
increase of atmospheric $CO_2$ concentration (Dufresne and Bony, 2008).
Recent remote sensing studies have shown that the clean and stable arctic atmosphere is
characterized by a high occurrence of mixed phase clouds (MPC) all year long, except in winter
and early spring when ice clouds are important (Mioche et al., 2015). However, the Svalbard
region is an exception where MPC are the most frequent cloud independent of season (Mioche
et al., 2015). Moreover, the altitude of the MPC is highly dependent of the height of the
inversion layer. The frequently occurring situation with a stable atmosphere and the low level
pronounced inversion layer promotes low level clouds of stratus form (Curry et al., 1996). The
arctic MPC are composed of a liquid layer on top and below which is located the mixed phase
where ice crystals take form (Gayet et al., 2009). If the crystals grow enough, a precipitation
layer is produced below the cloud. The dynamics together with possible surface coupling and
advection are essential to maintain the MPC during several days (Morrison et al., 2012). This
structure results from a complex network of interactions between numerous local and larger
scale processes that complicates the understanding of the MPC properties evolution and its
impact on arctic climate (Morrison et al., 2012). In the Arctic, the umbrella effect is not
necessarily dominant compared to the cloud greenhouse effect (Quinn et al., 2008), which
suspects that clouds play an important role in the arctic amplification. Several studies have
revealed that MPCs have a large impact on the surface radiative flux in the Arctic (e.g. Kay et
al., 2012, Wendisch et al., 2013).
Arctic cloud properties are linked to aerosol properties since they can act as Cloud
Condensation Nuclei (CCN) or Ice Nuclei (IN). Thus, aerosol seasonal variability and transport
from lower latitudes play a role in cloud properties evolution. Studies have shown an arctic
annual mean aerosol concentration half that for mid-latitudes. The stable atmosphere and the
dark winter promote growth by coagulation/coalescence of the particles, i.e. an increase in size
and decrease in concentration, with dominant accumulation mode (Tunved et al., 2013). When
the sun rises during spring, these big particles, which can come from lower latitudes, generate
the arctic haze phenomenon (Quinn et al. 2007). The stronger sun light gives rise to increasing
photochemical activity associated with new particle formation and a dominant Aitken mode
(Engwall et al., 2008). Moreover, the ice melting exposes land surfaces that can act as aerosol
sources. Therefore, the aerosol concentration increases until its maximum in summer. These
features were observed in Alaska (Quinn et al., 2002) and Svalbard (Tunved et al., 2013),
proving that they are representatives of the aerosol properties evolution in the Arctic.



The rapid change in aerosol properties occurring in spring is known to cause changes in arctic
cloud properties, the so-called aerosol indirect effect. Increase in aerosol concentration with
constant Liquid Water Path (*LWP*) is known to increase cloud droplet concentration and cloud
optical thickness but decrease droplet size (Twomey, 1974, 1977), decrease the precipitation
efficiency and increase the cloud lifetime (Albrecht, 1989). Also, in a temperature rise scenario,
the cloud height is expected to increase (Pincus and Baker, 1994). The impacts of anthropogenic
aerosol transported to the Arctic on clouds are not fully understood, but Garrett and Zhao (2006)
showed that the cloud emissivity is higher for polluted case, contributing to the arctic warming.
In the case of artic MPC where liquid and ice phases coexist, the aerosol-cloud interaction is
complexified by the addition of the ice phase and several interaction mechanisms have been
assumed. Lohmann (2002a, 2002b) proposed that an increase in ice nuclei could increase the
cloud ice content at the expense of the liquid content. This so-called glaciation indirect effect
would mean, as the precipitation is more efficient for the ice phase, a decrease in cloud cover
in lifetime. The riming indirect effect predicts a riming efficiency decrease due to the
supercooled droplet size decrease. Thus, an increase in Cloud Condensation Nuclei (CCN)
could lead to a decrease in Ice Water Content (IWC) and ice particles concentration (Borys et
al., 2003). According to the data of the two measurement campaigns ISDAC (Indirect and Semi-
Direct Aerosol Campaign) and MPACE (Mixed-Phase Arctic Cloud Experiments), Jackson et al.
(2012) found a correlation corresponding to the glaciation effect above the cloud liquid phase
but no evidence of the riming effect. Mc Farquhar et al. (2011) showed that the aerosol size is
the main parameter to explain the particles activation and that the chemical properties don't
determine the ability of an aerosol to act as a CCN, i.e. the Kelvin effect is dominant compared
to the Raoult effect in the Arctic.
The work presented here is included in the frame of the project CLIMSLIP (CLimate IMpacts
of Short-LIved Pollutants in the polar region). The main objective of this project is to reduce the
uncertainties of the radiative forcing due to the anthropogenic emissions of tropospheric ozone,
methane and aerosol including Black Carbon (BC). This article will focus on the arctic ground
based in situ cloud and aerosol measurement study, performed at the Mount Zeppelin station
(474 meters altitude), in Ny-Alesund, Svalbard, performed during spring 2012. First, a
classification and characterization of the different types of cases will be presented. Then, a
comparison between a polluted and a clean case will be made, based on air masses
backtrajectories. In the end, the different aerosol-cloud interactions will be discussed and, if
possible, quantified.

## *2 Site & instrumentation*

### 2.1 Site

The campaign was carried out between February 26 and May 2 at the Mount Zeppelin station
(78°56'N, 11°53'E) located south-west of the Ny-Alesund village, Svalbard, at an altitude of 474
meters above sea level. This station presented in Figure 1 was built and is managed by the
Norwegian Institute for Air Research (NILU). The Zeppelin observatory is mostly unaffected by
local sources and is considered to be within the boundary layer most of the time (Tunved et al.,
2013). This station represents remote arctic conditions and is a part of the European observation
network ACTRIS (Aerosols, Clouds, and Trace gases Research InfraStructure network).
Continuous measurements of atmospheric trace gases and aerosol physical and chemical properties



are performed all year long. The station is also equipped with instruments to measure temperature,
humidity and wind speed and direction.
A ceilometer, CL51 model, was installed in the Ny-Alesund village at sea level. This remote
sensing instrument is designed to measure the clouds within an altitude range between 0 and 15
km. It uses the technology of a lidar with a laser wavelength at 910 nm. During CLIMSLIP, the
ceilometer was used to retrieve the approximate altitude of the mixed phase and the liquid layers
and showed good agreement with the microphysical measurements. However, in some cases,
fog or an optically thick ice layer prevents the laser beam from penetrating within the cloud
system.

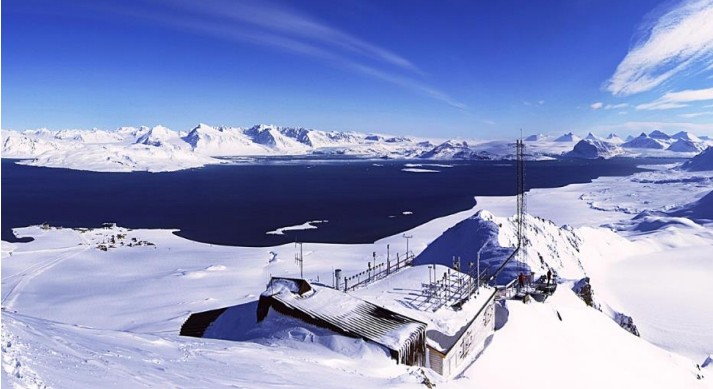

155                   Figure 1: Picture of the Mount Zeppelin station (www.npolar.no)

## 2.2 Instrumentation and data processing



2.2.1 Cloud instrumentation

The cloud ground based instrumentation used during CLIMSLIP-NyA was installed on a
measurement pole and is presented in Figure 2. The cloud optical and microphysical properties
were thus assessed by three independent instruments: a PMS Forward Scattering Spectrometer
Probe (FSSP-100), a Cloud Particle Imager (CPI) and a Polar Nephelometer (PN). They were
all connected to the same pump by plastic tubes, leading to the sampling volume indicated on
Figure 2. They were operated approximately 2 m above the platform level and mounted on a
tilting and rotating mast, allowing them to be moved manually in the prevailing wind direction.
The proper alignment of their inlet with the flow was based on the wind direction measurements
performed by a mechanical and ultrasonic anemometer.

The FSSP-100 measures the number and the size of particles going through the sampling
volume, from the forward scattering of a 632.8 nm wavelength laser beam (Knollenberg, 1981,
Dye and Baumgardner, 1984). Using the Mie theory, this instrument is dedicated to droplets.
The Particle Size Distribution (PSD) is thus computed in 15 adjustable size classes with
uncertainties on the effective diameter and *LWC* of respectively 2 µm and 30 % (Febvre et al.,
176    2012).
The CPI is an imager and takes pictures of the particles when going through the detection
volume with 256 grey levels, thanks to a CCD camera with a resolution of $1024 \times 1024$ pixels.
These images allow computing the particles size and so the PSD, but several morphological




parameters are also retrieved and are used to classify the sampled particle in 10 shape
categories: spheroid, needle, column, plate, bullet, stellar, graupel, rosette, sideplane and
irregular (Lefèvre, 2007). However, a manual classification has been done during the
CLIMSLIP campaign due to some malfunctions of the automatic classification. The
determination of the IWC is realized according to the Baker and Lawson (2006) and Lawson
and Baker (2006) method. The uncertainties on the concentration and the effective diameter are
assessed respectively as 50 % and 80 %.
The PN measures the scattering phase function of a set of cloud particles thanks to a 804 nm
wavelength laser beam and 56 photodiodes distributed over scattering angles between 3.5° and
172.5° (Gayet et al., 1997). From the scattering phase function can be computed two important
integrated optical parameters, the extinction coefficient and the asymmetry parameter with
accuracies estimated within 25% and 4%, respectively (Gayet et al., 2002).
The Nevzorov probe is a hot wire device at constant temperature with two captors and an
electrical resistor. The particles are vaporized, and an electrical power is provided to the
resistor. The resulting power is relied related to the *LWC* and *TWC*, depending on the captor
(Korolev et al., 1998). Due to high discrepancies, this instrument was used only for instrumental
comparison and data processing analysis and will not be discussed further.

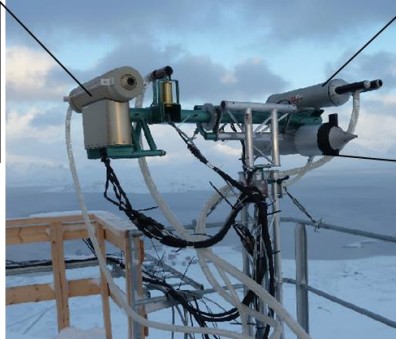

Figure 2: Cloud instrumentation used during CLIMSLIP. Indicated are: particle size range,
main cloud properties measured and theoretical sampling speed.
2.2.2 Aerosol instrumentation
The particle inlet at the Mount Zeppelin station is a Whole Air Inlet, which possesses a heating
system that prevents the inlet to be filled by ice or frost and to evaporate the condensed water
or ice. Thus, all the aerosols (CCN, IN or interstitial) are sampled by the instruments described
hereafter. The aerosol sampling covers particles sizes between 3 and 809 nm (Tunved et al.,
209  2013).
The Mount Zeppelin aerosol instrumentation is composed of one Condensation Particle Counter
(CPC), one Differential Mobility Particle Sizer (DMPS), one aethalometer and one aerosol
nephelometer, which are running continuously throughout the year. The CPC, 3015A model, is
a particle counter for aerosol diameters larger than 3 nm. It measures aerosol concentration up
to $10^5$ particles/cm$^3$ with an accuracy of 10 % (TSI, 2002). The DMPS is a CPC combined with
a Differential Mobility Analyzer (DMA), which allows selecting different size ranges. The
aerosol PSD is obtained with 22 diameter classes going from 25 to 809 nm. The aethalometer
assesses the Black Carbon (BC) concentration based to the optical extinction of the aerosols



collected on a filter (see Eleftheriadis et al., 2009, for details). The nephelometer measures the
aerosol scattering coefficient for three wavelength: 450, 550 and 700 nm (TSI, 2005). During
CLIMSLIP, this nephelometer was used with a time resolution of 5 minutes.
2.2.3 Data processing
The three cloud instruments operated at a one Hz resolution. The data processing has followed
the conclusions of the cloud instrumentation study presented in Guyot et al. (2015). This paper
highlights the biases that can exist between the instruments and the need of an Ensemble of
Particles Probe (EPP) to standardize the data. In the case of the CLIMSLIP campaign, such
correction was not possible for two reasons not developed further. (1) Strong discrepancies of
the EPP Nezvorov probe, probably because of a too low sampling speed. (2) The
standardization according to the extinction coefficient of the PN is not consistent with the
aerosol data (there are more droplets than CCN). Thus, this study will not provide quantitative
results but qualitative ones based on case comparisons and variation studies.
According to Guyot el al. (2015), measurements with an angle between the instruments
orientation and the wind direction higher than 30° can modify the PSD due to changes in the
sampling conditions. Those measurements were therefore not taken into account for the study.
Moreover, the ground based low sampling speed induces low sampling rate, especially for the
CPI with values between 0.5 and 20 sampled particles per minute. This doesn't allow us to
work on low time resolution scale. To get sufficient particle statistics, the minimum average
time resolution will be 1 minute for the FSSP and one day for the CPI.
During the aircraft campaign, a cloud particle can break on impaction with the inlet due to the
high sampling speed corresponding to the plane speed. This results in more numerous and
smaller droplets or crystals and creates artifacts in the PSD (Rogers et al., 2006). Due to the
low sampling speed, ground based measurements has the advantage to avoid this effect, but at
the expense of the sampling rate.

## *3 Identification and characterization of the study cases*


## **3.1 Overview**

Several kinds of episodes were met during CLIMSLIP. Figure 3.a shows those episodes with
the time series of measured temperature and relative humidity. Thus, according to the
ceilometer measurements and observations, we enumerate:
-  4 episodes of sampling of the liquid and mixed phase layer (LMPL) of MPC, on March
11th and 29th and April 27th and 28th.
-  3 cases of sampling of the precipitation layer of MPC, on March 28th and April 14th and
20th.
-  2 occurrences of Blowing Snow (BS), on March 23th and 31th.



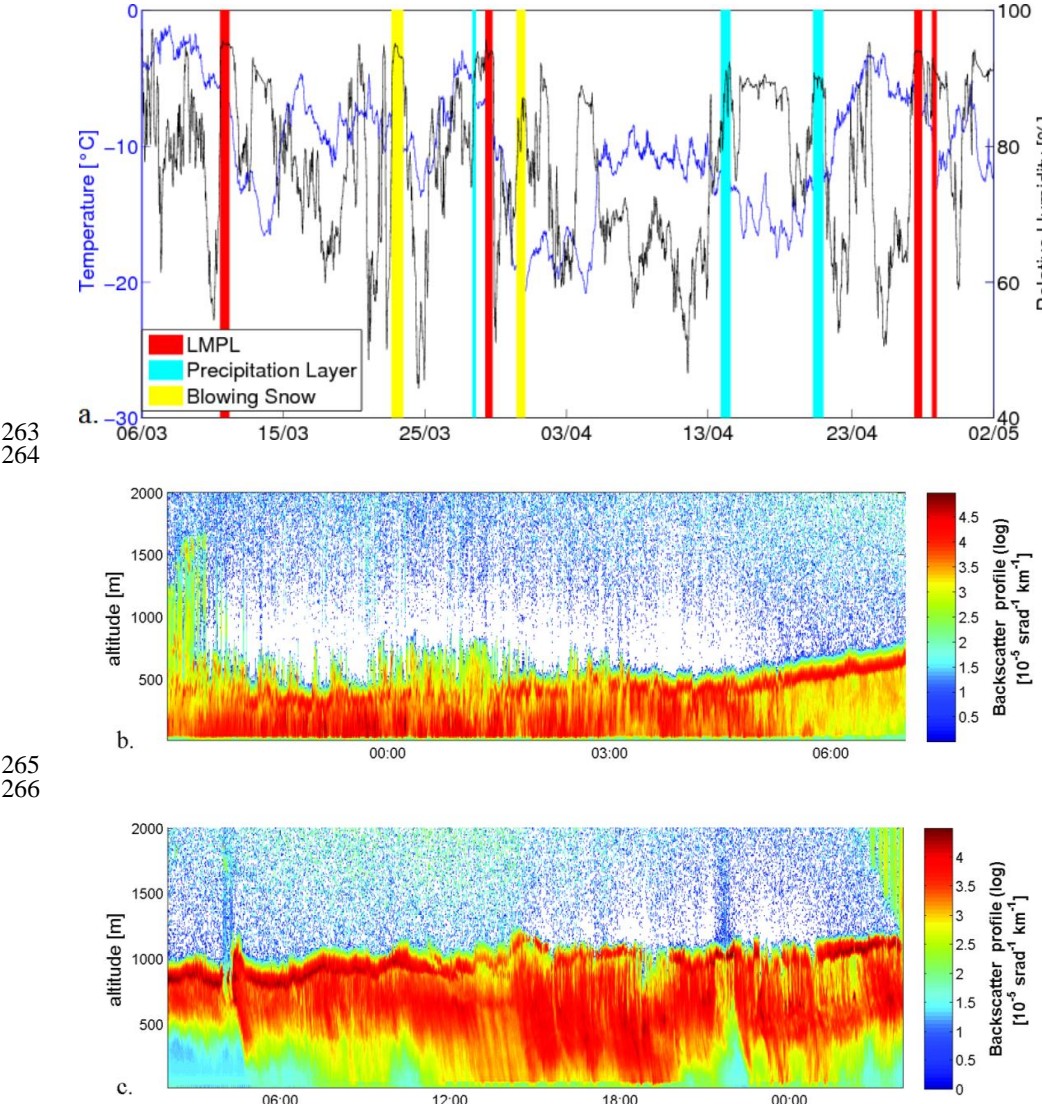



Figure 3: a) Time series of the temperature and relative humidity measured at the Zeppelin
station. The different cases are plotted with colored columns. Examples of time series of the
ceilometer attenuated backscattered coefficient profile in km$^{-1}$ sr$^{-1}$ for b) MPC case (April
27$^{th}$) and c) precipitation case (April 14$^{th}$).

The cases called "LMPL" and "Precipitation layer" both reveal the presence of a MPC, i.e.
where cloud in situ instrumentation sampled ice and/or liquid particles. But, in the first case,
the ceilometer shows the liquid layer around 500 meters altitude (Figure 3.b). This liquid layer
having a very strong extinction coefficient, the ceilometer beam does not go through, what
happens above the liquid layer is therefore unknown. On the same time, droplets are sampled
by the FSSP. Following the altitude of the cloud, the station is in the liquid layer or the mixed
phase layer. These episodes are characterized by relative humidity maximum.



In the second case, the ceilometer locates the liquid layer around 1 km altitude or more (Figure
3.c). No droplets are sampled. The station is so below the mixed layer, within the ice
precipitation. This layer has a variable extinction coefficient depending on the crystal density
but the laser beam is not completely attenuated. The relative humidity shows high values around
90 % but remains lower than the MPC cases.
Moreover, the temperature varied between -20 to -1 °C, so it remains always below the
solidification point, liquid particles were always supercooled droplets. The Blowing Snow
episodes will be discussed in annex.
In the following, the LMPL and precipitation layer cases will be microphysically and optically
characterized. These characterizations will be useful to determine futures measurements that
are not completed with visual observations (e.g., remote sensing measurements). Moreover,
combined with other measurement campaign in the Arctic, we hope to increase knowledge
about growth processes in low level mixed phase arctic clouds.

## 3.2 Characterization of the LMPL cases

Arctic MPC can be characterized by a succession of layers with liquid or ice dominance. The
phase heterogeneity is both horizontal and vertical. Because of the fixed position of the
measuring station, we could not control the location of the measurements within the cloud
system. However, a characterization of the mean parameters is possible.
The determination of the thermodynamic phase of a cloud can be based on microphysical and
optical criteria. Figure 4 presents the occurrence number of the MPC liquid fraction $F_{liq}$ and the
asymmetry parameter g. $F_{liq}$ is computed as :

$$F_{liq} = \frac{LWC_{FSSP}}{(LWC_{FSSP} + IWC_{CPI})} \qquad (1)$$

The results show a higher observation frequency for extreme $F_{liq}$ values (close to 0 or 100 %).
The minimum frequency is between 20 and 70 %. This means that the low level mixed phase
cloud layers are preferentially with liquid or ice dominance for the spatial resolution of our
measurements. This confirms the conclusions from the scientific literature (e.g., Gayet and al.,
2009; Korolev and Isaac, 2006).
Moreover, g shows a more or less linear relation with $F_{liq}$. This highlights the relation between
the optical properties and the microphysical properties. Therefore, the knowledge of the MPC
microphysical properties is a key parameter to reliably assess the radiative transfer in the Arctic.
The g variability is significantly larger for $F_{liq}$ below 50 %. This tends to show a more complex
optical behavior for ice dominating layers.





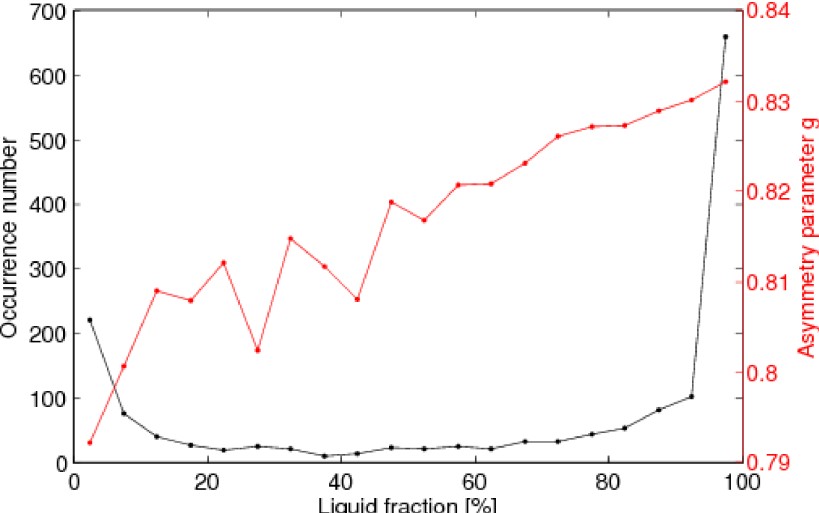

Figure 4: Occurrence number and mean values of g in relation to the liquid fraction $F_{liq}$ for the
323        four LMPL cases. $F_{liq}$ is derived from the CPI and FSSP measurements (see Eq. 1) with 1
324            minute resolution corresponding to a spatial resolution of 800 meters.

Figure 5 shows the average PSD, from 3 µm to 2.3 mm, obtained with the FSSP and the CPI
for the four LMPL cases. The mean $F_{liq}$ is also indicated. The four PSD show similar trends,
i.e. two modes centered at 10 µm for droplets and around 250 µm for ice crystals.
According to Costa et al. (2014), these PSD correspond to the coexistence regime characterized
by $RH_w$ (relative humidity according to liquid water) and $RH_i$ (relative humidity according to
ice) > 100 % and stable coexistence of crystals and supercooled liquid droplets with the droplet
PSD $10^6$ higher than the crystal PSD. This is opposite to the Bergeron regime where $RH_w$ < 100
% and $RH_i$ > 100 %, so the crystals grow in expense of the droplets (Costa et al., 2014). This
reveals that the Wegener-Bergeron-Findeisen process doesn't alone explain the formation and
growth of ice crystals.
However, the March 11$^{th}$ and 29$^{th}$ PSDs show differences with the other cases with a high
concentration for the smallest CPI classes. This is due to big droplets sampled by the CPI. The
FSSP doesn't show such consequent differences in droplet PSD or diameter. We also point out
that the absolute values should not be taken into account. Indeed, in addition to instrumental
issues (see Guyot el al., 2015), the results and the differences between the cases are largely
dependent on the station residence time within the liquid or mixed layer which cannot be
controlled. Similar PSDs were observed at the Mount-Zeppelin station by Uchiyama et al.
(2013) in 2011. This publication concludes that the liquid/ice distribution is a function of the
cloud evolution stage; we highlight here the importance of the station position inside the cloud
system for our data analysis.


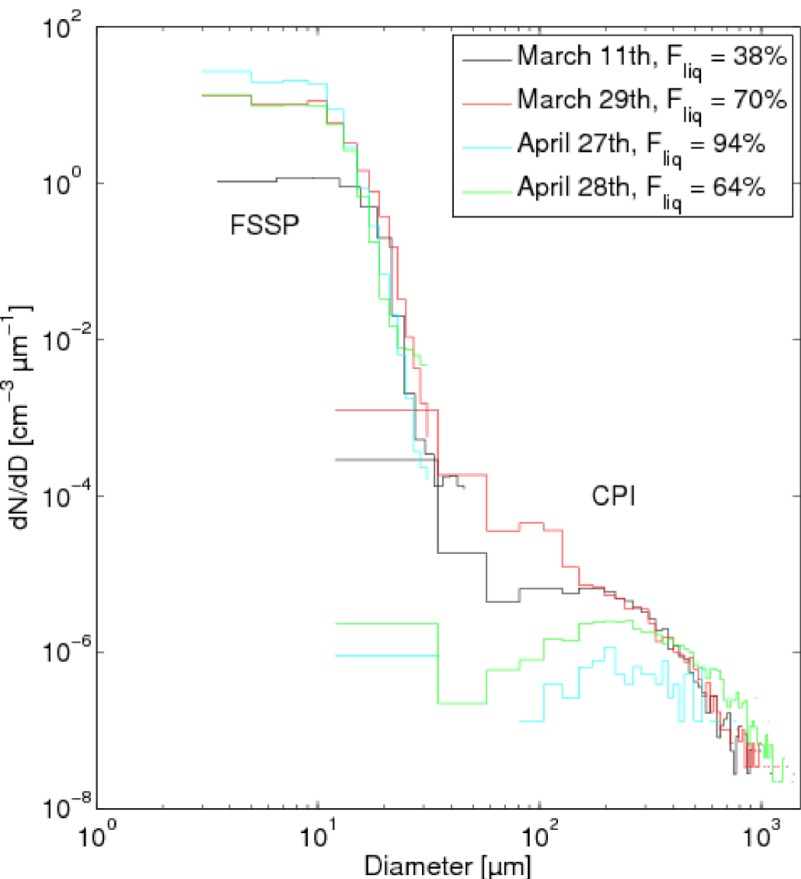

Figure 5: Cloud PSD in cm$^{-3}$ µm$^{-1}$ measured by the FSSP [3-50 µm] and the CPI [15-2300
µm] and average for the four LMPL experiments. The mean values of $F_{liq}$ are indicated in
legend.
The shape classification performed by the CPI is presented Figure 6.a. The high droplet
concentrations of the smallest CPI classes observed on March 11$^{th}$ and 29$^{th}$ (see Figure 5) are
responsible of the strong number dominance of the droplets with a value of around 85 %.
However, liquid water represents a very small proportion in mass and surface fractions. For
these two quantities, side planes and irregular shapes dominate.
The assessment of the crystal growth mode is confronted to the fact that the measurement
station can change its position in the cloud. An evolution in the CPI PSD is so not necessary
due to particles growth. However, the crystal shape, accurately measured by the CPI, is a good
indicator for the growth mode and the high percentage of regular shape would indicate a growth
dominated by vapor deposition. Aircraft measurements performed in Svalbard in 2007 show
similar results, with in particular strong presence of irregulars and side planes for altitudes and
temperatures around 500 m and -12 °C, respectively (Gayet et al., 2009)




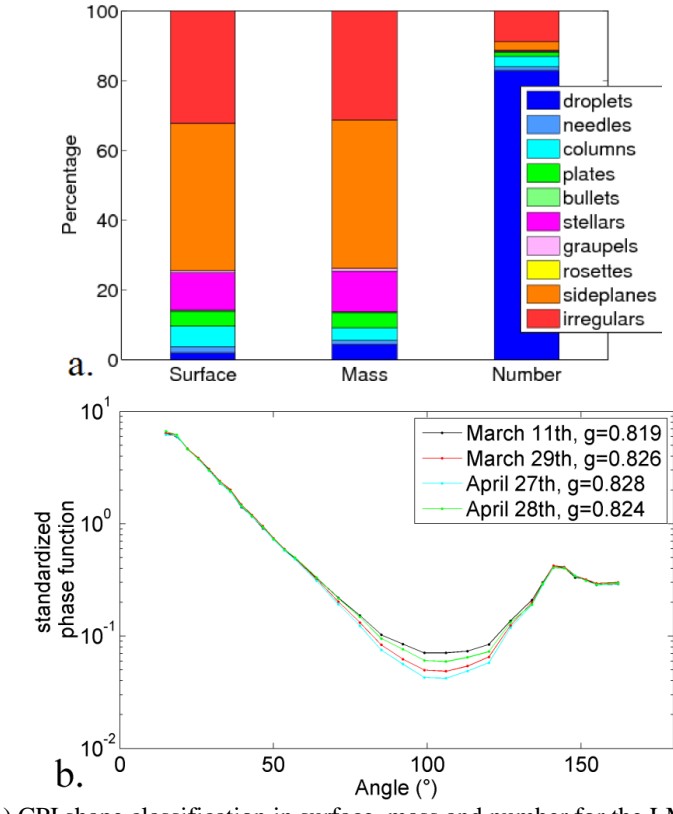

Figure 6: a) CPI shape classification in surface, mass and number for the LMPL cases. The
color represents the occurrence percentage of the shapes as indicated by caption, and b)
average standardized phase functions measured by the PN for the four LMPL cases. Caption
indicates the asymmetry parameter.
Measurements of the cloud particle scattering properties performed by the PN allow to study
the optical signature of the main microphysical properties observed on the MPC. Figure 6.b
displays the average phase functions and asymmetry parameters ($g$) for the four LMPL cases.
Differences between the experiments are negligible in forward and backward scattering but
within the lateral scattering domain [60°; 130°]. The scattering increases when g decreases. The
1 minute average $g$ values during the whole measurement campaign are included between 0.74
and 0.85, which is consistent with results obtained by Garrett et al. (2001).
Combined with Figure 5, these results show that $g$ and the lateral scattering are relied to
microphysical properties. Indeed, lateral diffusion increases when $F_{liq}$ decreases. Therefore,
March 11[th] experiment presents the lower $F_{liq}$ (38 %), the higher lateral scattering and the lower
average value of g (0.819). The contrary is shown in the April 27[th] case ($F_{liq}$ = 94 %, $g$ = 0.828).
This is consistent with previous studies (Gayet et al., 2009; Jourdan et al., 2010) and also proves
the qualitative coherence between the FSSP and the PN.
The analysis of the optical-microphysics coupling is limited by the sampling speed and rate and
the PN measurement accuracy. Indeed, a mean component analysis failed to establish a
relationship between the phase function and the crystal morphology, as highlighted by Jourdan
et al. (2010).
**3.3 Characterization of the precipitation cases**

Figure 7 displays the average CPI PSD for the three cases of precipitation. The FSSP is blind
for those particles sizes. The April 14[th] and 20[th] experiments show a PSD with very low
concentrations, close to the detection limit, centered around 350 µm and accompanied by
relatively low temperature < -10°C. The March 28[th] case differs from the two other experiments
with higher concentrations and a PSD centered around 200 µm, similar to the LMPL cases.
Besides, the temperature is higher with an average value of -5 °C. This could reveal an influence
of the mixed layer and/or temperature effect.
However, the ceilometer located the cloud base at an altitude of approximately 1000 m for the
three days, which would indicate that the station position doesn't explain the differences. The
temperature differences could lead to different growth processes and so different sizes.
This information can be provided by the CPI image classification presented Figure 8.a showing
a pronounced presence of stellars. Even if the stellar crystals aren't a majority in number, they
stand for more than half of total surface and mass. However, the number shape distribution was
not identical for the three days. Indeed, the 14[th] and 20[th] April experiments are dominated by
stellar whereas the March 28[th] case shows a much more important contribution of plaque,
irregular and needle. As the concentration is 5 times higher for the March 28[th] case, its
contribution in the total number distribution is more important.
Therefore, even if the temperature measured at the station is potentially different than the crystal
formation and growth temperatures and the oversaturation according to the ice was not
measured, we have seen that the CPI measurements show that temperatures below -10°C are
favorable to the formation of big size crystals such as stellar, whereas, for warmer temperatures,
plaque, irregular and needle crystals with smaller sizes dominate. This agrees with the crystal
classification studied by Bailey and Halley (2009) and explains the differences in the daily PSD.


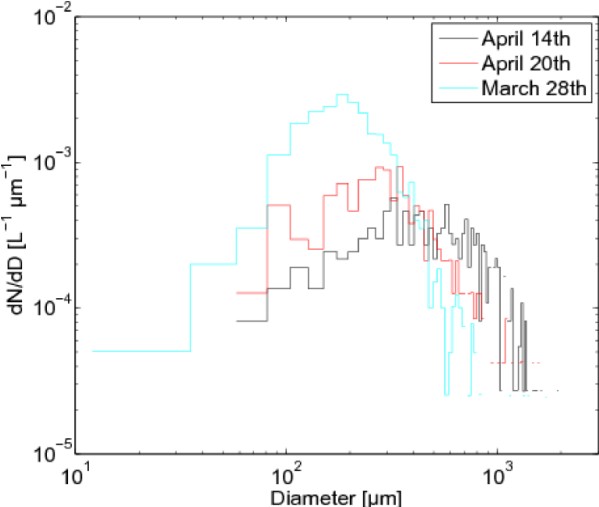

Figure 7: Same as Figure 5 for the precipitation cases.





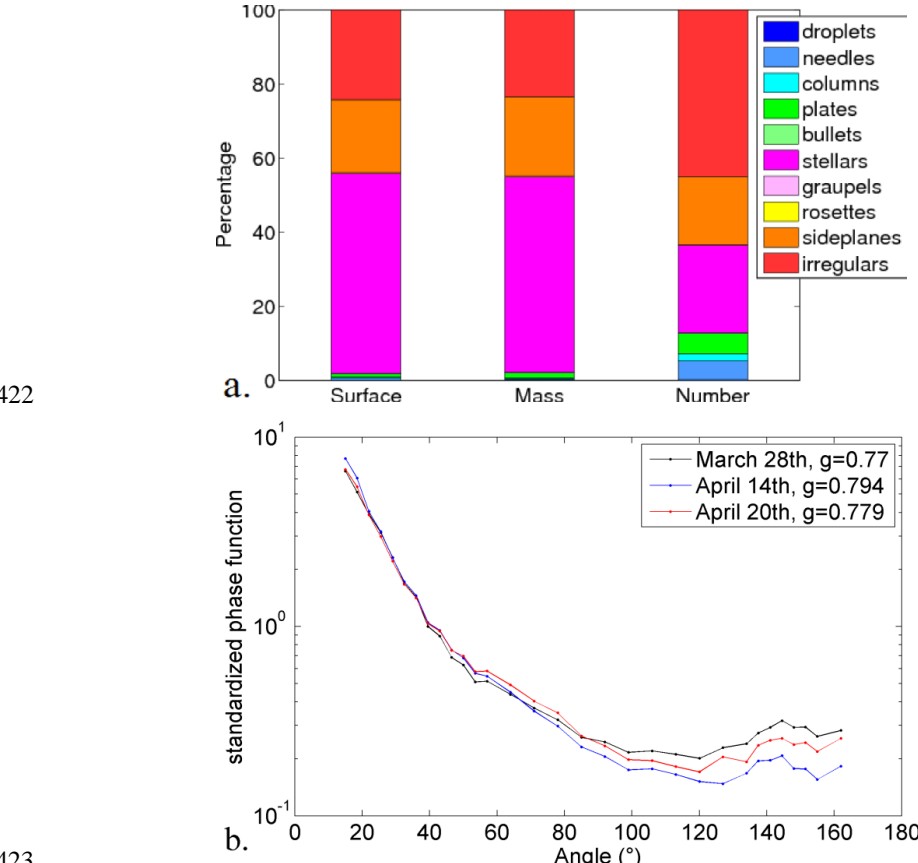


Figure 8: Same as Figure 6 (a and b) for the precipitation cases.
Figure 8.b presents the average phase functions of the precipitation cases. The lateral scattering
is more important than for the measurements of the mixed and liquid layers. The asymmetry
parameter is lower around 0.79 which is typical for ice particles. Notable differences are
observed for scattering angles as low as 40°. These differences are probably due to the crystal
morphology variability. Unfortunately, such relationships were not observed with the
CLIMSLIP data. Indeed, a principal component analysis didn't allow discriminating the phase
function according to the crystal shape. This can be explained by the low sampling rate during
the precipitation events involving a very low crystal statistics.
To conclude, the results were limited by the low particle sampling rate and the uncontrolled
position of the station inside the cloud system. However, differences between LMPL and
precipitation layer have been explicated and allow a quick recognition without visual
observations in future studies. These results will be compared to other measurement campaign
for a better understanding of the microphysical processes and feedbacks that take place in low
level mixed phase arctic clouds.



## 4 Aerosol-cloud interaction in the Arctic

The objective of this part is to quantify the effects of the aerosol properties on the cloud properties observed during the CLIMSLIP campaign. To do this, we will in a first step compare the two experiments of March 11th and 29th that will be the "clean" and "polluted" cases, respectively. In a second step, several aerosol cloud-interaction processes will be evaluated and in situ measurements will be used to assess quantities that are required in parametrization of the arctic aerosol-cloud interaction.

### 4.1 Section on tools: the FLEXPART-WRF model and definitions

This analysis will be supported by results from the lagrangian particle dispersion model FLEXPART-WRF (version 3.1, Brioude et al., 2013) adapted from the FLEXPART model (version 6.2, Stohl et al., 2005). FLEXPART-WRF simulates long distance transport and, in a mesoscale, the moist and dry scavenging and the diffusion of atmospheric tracers and air masses (see Stohl et al., 1998, Stohl and Thomson, 1999, or Stohl et al., 2005, for more details). The FLEXPART-WRF model was driven by WRF (Skamarock et al., 2008) meteorological forecasts to provide air masses back-trajectories and several tracers' origins.

For each single run, 20000 pseudo-particles were released from a small volume surrounding the analyzed position. Then, they were then tracked backward in time. The model output a tridimensional distribution of the Potential Emission Sensitivity (PES) on a 1° longitude x 1° latitude resolution grid. The PES is expressed in s/kg, which corresponds to the residence time of air particles within a given cell. In order to investigate the potential sources of the pollution transported to the Arctic and since the pollutants generally remain below the inversion layer, the model output is integrated over the first kilometer atmospheric column and becomes a Footprint PES (FPES). Combined with the ECLIPSE (Evaluating the Climate and Air Quality Impacts of Short-Lived Pollutants, see Klimont et al., 2016) atmospheric pollutants emission inventory, FLEXPART-WRF provides a valuable insight on the potential geographic contribution of anthropogenic sources for pollution tracers such as CO, $SO_2$ and BC. The combination between the FPES and the emissions is called the Potential Source Contribution (PSC) expressed in kg of tracer per air kg. In this study, we will focus on the CO tracer which gives an assessment on the origin of the anthropogenic pollution transported to Svalbard.

The aerosol cloud interaction study will also be supported by two other parameters: the activation fraction $F_a$ and the activation diameter $D_a$. $F_a$ can be defined as the percentage of aerosols becoming CCN (Abdul-Razzak et al., 1998) and is computed by the ratio of the FSSP and CPC concentration. The CPC was chosen because it provides the largest aerosol size range:

$$F_a = \frac{N_{drop}}{N_{aerosol}} = \frac{N_{FSSP}}{N_{CPC}} \qquad (2)$$

We define $D_a$ as the diameter beyond which all the aerosols are activated, assuming the aerosol chemistry effect is negligible (Abdul-Razzak et al., 1998). During CLIMSLIP, $D_a$ is calculated as the DMPS diameter for which the DMPS total concentration is equal to the FSSP concentration. Even if the aerosol size range is smaller for the DMPS than the CPC, the DMPS was chosen because the aerosol PSD is necessary:





$$\sum_{D_{max}}^{D_a} n_{DMPS}(D) = N_{FSSP} \qquad (3)$$

## 4.1 The "Clean" case of March 11th

The March 11th case, just like March 29th, presents a stable atmosphere with a low level mixed phase cloud. The liquid layer was sampled but, unfortunately, the ceilometer beam was almost entirely attenuated within the first 500 meters, avoiding to assess cloud top and base altitude. The sounding balloon show the inversion layer around 925 mb (700 m) for both days.

Figure 9 shows the time evolution of the DMPS, CPC and FSSP concentration, the activation fraction and the average BC concentration. The DMPS ceased to work from 7:30 until the following day, but, as the CPC and aethalometer parameters show almost constant values until 12:00, the DMPS concentration is assumed to do the same. The DMPS concentration is plotted for different particles sizes (total, > 50 nm and > 100 nm). The DMPS PSD shows a bimodal distribution with a pronounced Aitken mode which is as important as the accumulation mode (not shown). This is obvious in Figure 9 where the accumulation mode concentration, i.e. particles sizes larger than 100 nm, equals half the total concentration.

The CPC displays an aerosol concentration (> 3 nm) relatively stable and weak between 100 and 130 cm$^{-3}$. The average BC concentration reaches 22.6 ng m$^{-3}$ during the liquid episode. The FSSP shows a droplet concentration up to 100 cm$^{-3}$, which leads to $F_a$ values between 60 and 80 % for the sections clearly in the densest zone of the MPC liquid layer.

The FSSP droplet effective diameter is around 12 µm and the DMPS effective diameter around 250 µm. $D_a$ shows very high variations with a mean value around 150 nm (not shown).

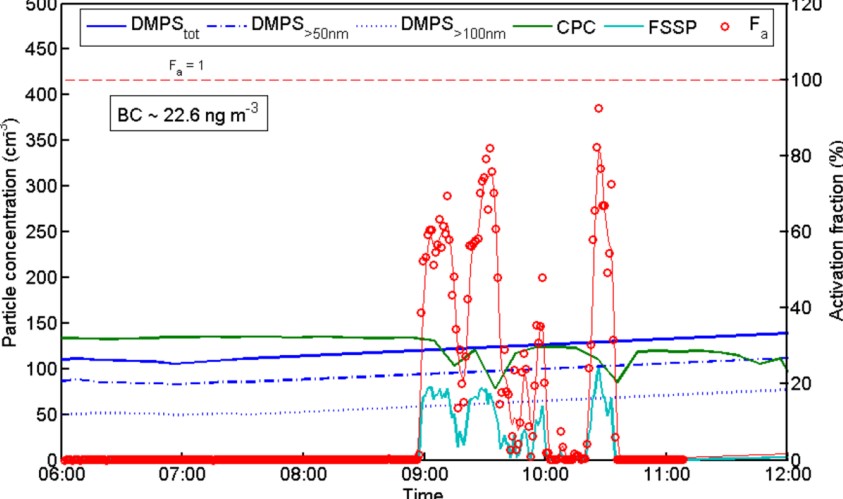

Figure 9: Time series of aerosol concentrations measured by the DMPS and the CPC, droplet FSSP concentration and the activation fraction, for the March 11th. The DMPS concentration was divided into three groups: the total concentration [25 - 809 nm], particles larger than 50 nm [50 – 809 nm] and larger than 100 nm [100 – 809 nm], the latter corresponding to the accumulation mode concentration. The activation fraction has been plotted with a sliding average of 5 minutes; the average aethalomètre BC concentration is indicated.





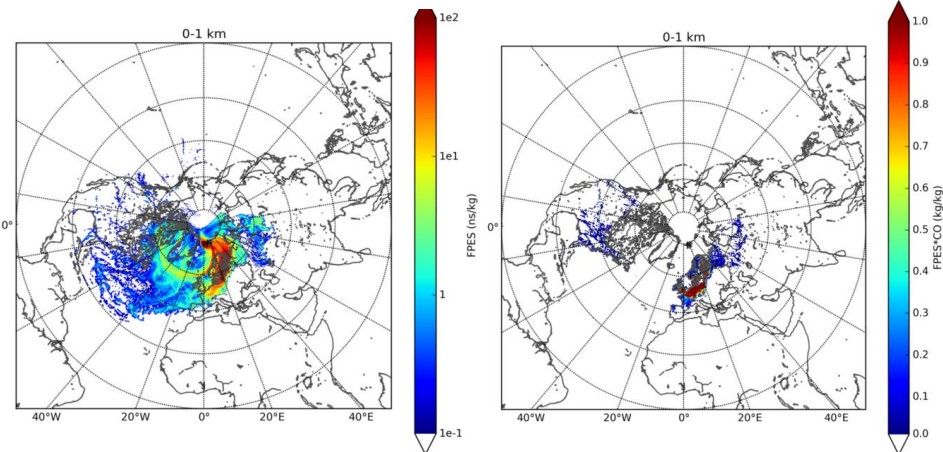

Figure 10: a) FLEXPART-WRF 12 days FPES from the simulation initiated from Mount-
Zeppelin on March 11[th] between 9 AM and 12 AM UTC. b) PSC computed from the FPES of
526                   Figure a) for the CO, expressed in kg CO per kg air.
Figures 10 a and b present respectively the FPES over 12 days and the PSC of CO for the air
mass arriving at the station during the liquid episode of the March 11[th] described in Figure 9.
The FPES shows that the aerosol sources are mainly located in the north of Scandinavia and so
that the long-range transport of anthropogenic aerosols is relatively limited. Indeed, over the
FLEXPART-WRF time computation of 12 days, the air masses come principally from Svalbard
and Scandinavia surrounding, showing very slow move. The CO PSC map presents an
anthropogenic origin dominated by North Europa: Scandinavia, north of Germany, Netherland,
Belgium and north of France.
The closer air masses origin makes this case the "clean" case. The important contribution of
local aerosol sources, mainly composed of gaseous precursors for the arctic region during this
period of the year (Quinn et al., 2007), explains the relative small aerosol mean diameter and
the high Aitken mode concentration observed by the DMPS (see Figure 9).

**4.2 The "Polluted" case of March 29[th]**
Figure 11 displays the same time series as Figure 9 for the liquid episode of March 29[th]. The
CPC and DMPS total concentration are decreasing going respectively from 220 cm[-3] to 120 cm[-3]
and from 175 cm[-3] to 80 cm[-3], due to the scavenging by ice precipitation. The FSSP droplet
concentrating reaches 150 cm[-3] and the average BC concentration 65.8 ng m[-3]. Comparing to
the March 11[th] case, these four concentration are all higher during the March 29[th]. The activation
fraction is also higher on March 29[th] with values between 80 and 100 % in the liquid layer and
$F_a$ increases as the aerosol concentration decreases.
Just like March 11[th], $D_a$ shows very variable values around 150 nm (not shown). Its high
variability makes this parameter unsuitable when comparing the two cases. However, $D_a$
decreases from around 150 nm to 50 nm and the FSSP effective diameter increases from 8 to
10 µm when $F_a$ increases, proving that smaller aerosol particles are activated and droplets grow
when the aerosol number decreases.


Moreover, the DMPS PSD shows that 90 % of the aerosol concentration is included in the
accumulation mode, with an effective diameter almost constant at 300 nm. Therefore, the
aerosol diameter is larger and the droplet diameter is smaller for the March 29th case compared
to the March 11th case.

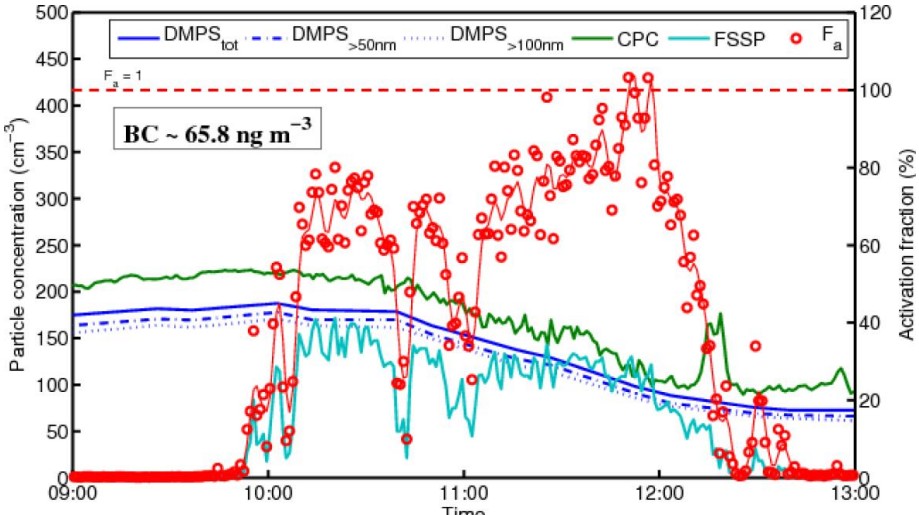

Figure 11: Same as Figure 9 for the March 29th case
The differences observed between the two days can be explained by the air masses origin.
Figure 12 shows the same FLEXPART-WRF FPES and CO PSC for the air mass arriving at
the station during the liquid episode of the March 29th. Backtrajectories distinguish clearly two
origin regions. The first one is Western Europa. The second air mass shows higher values of
time residence and comes from northeast Asia: northeast China and extreme east Russia. The
particularity of March 29th consists thus in this air mass coming from Asia which is the region
generally accepted to emit the highest aerosol concentration compared to the others regions of
the world (Boucher et al., 2013).
Therefore, compared to the "clean" case of March 11th, March 29th shows long range transport
of anthropogenically influenced air masses, leading to higher aerosol concentration in the Arctic
with especially a BC mass concentration 3 times higher. Thus, March 29th constitute the
"polluted" case. According to Quennehen et al. (2012), during the route, the Aitken mode
concentration quickly decreases by coagulation, for the benefit of the accumulation mode,
increasing the average effective diameter. This explains the accumulation mode dominance
observed in Figure 11 and the increase of the average DMPS effective diameter, and confirms
the strong influence of the lower latitudes emissions during the "polluted" case. On the contrary,
the "clean" case shows local sources composed of fresh particles, for at least half the
concentration.
This long range anthropogenic pollution has also strong influence on cloud properties. Indeed,
CCN abilities being mainly due to the aerosol size in the Arctic (Mc Farquhar et al., 2011),
accumulation mode dominance leads to higher aerosol effective diameter and higher $F_a$ values.
Combined with higher aerosol concentration, the droplet concentration increases whereas the
droplet size decreases meaning, theoretically, that the cloud optical thickness increases.





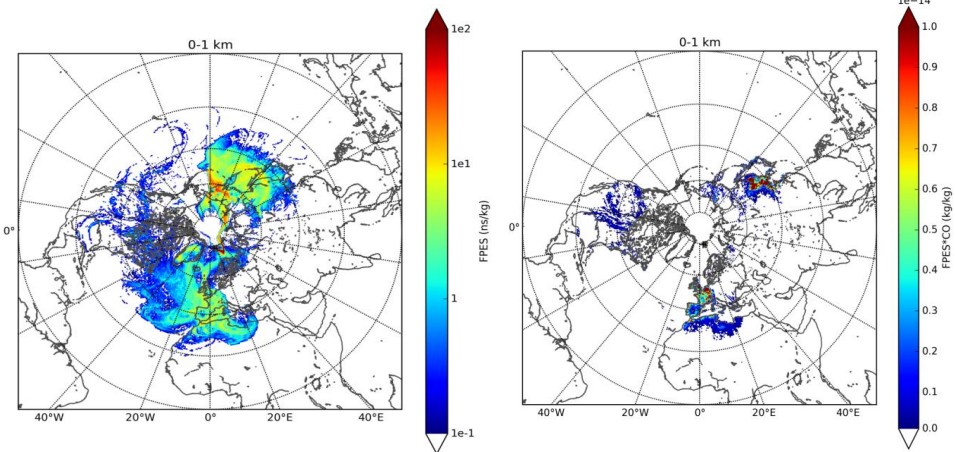

Figure 12: a) and b) Same as Figure 10 for the air masses arriving in the Mount-Zeppelin
station on March 29th between 10 AM and 1 PM UTC.
This qualitative study has to be completed with quantitative parameters that can be found in the
scientific literature. Therefore, the next section will focus on the quantitative variations of
droplet concentration and size according to aerosol properties. Moreover, glaciation and riming
indirect effect will be assessed.
**4.3 Quantification of the impacts of the aerosol properties on the cloud**
**microphysical properties**
The sensitivity of cloud diameter and concentration according to aerosol haze will be assessed
from two parameters, called the Indirect Effect parameter (*IE*) and the Nucleation Efficiency
(*NE*) and defined as follows (Feingold et al., 2001, 2003, Garrett et al., 2004):
$$IE = -\frac{\partial \ln(r_e)}{\partial \ln(\sigma)} \qquad (4)$$
$$NE = \frac{\partial \ln(N)}{\partial \ln(\sigma)} \qquad (5)$$
where $r_e$ is the droplet effective radius, $N$ the droplet concentration and $\sigma$ the aerosol scattering
coefficient.
We made two assumptions to use these parameters. First, *IE* and *NE* are assumed to evaluate
the variations of the droplet concentration and size according to the CCN concentration. To
measure this one, we use the scattering coefficient which is assumed to be proportional to the
CCN concentration. The accumulation mode particles are the most inclined to serve as CCN
because of their size and possess the highest scattering cross section compared to the other
modes (Garrett et al., 2004). Second, $r_e$ and $N$ are also dependent on the *LWP*, so *IE* and *NE*
have to be computed for similar *LWP* clouds (Feingold et al., 2001). During CLIMSLIP, the
*LWP* was not measured and we assumed that the *LWP* is effectively constant. This is reasonable
since the sampled clouds were all low level mixed phase arctic clouds and from the same season.


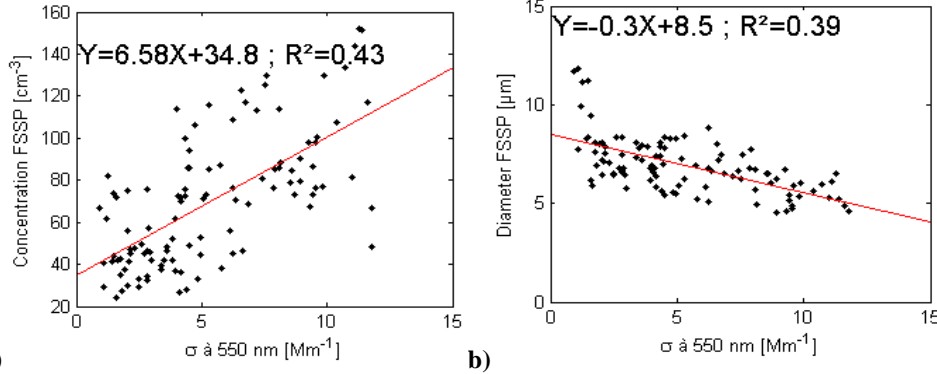

**a)**                                                    **b)**
Figure 13: Comparison of the aerosol scattering coefficient $\sigma$ at $\lambda = 550$ nm measured by the
nephelometer, with a) the droplet concentration, and b), the effective diameter measured by
the FSSP, for the four LMPL cases. Values of *LWC* below 5 10$^{-3}$ g m$^{-3}$ were not taken into
account. This comparison has been performed with a nephelometer time resolution of 5
minutes.
Figure 13 presents the comparisons between the droplet concentration and diameter with the
aerosol scattering coefficient. The results are consistent with the Twomey effect (Twomey,
1974, 1977) and the Albrecht effect (Albrecht, 1989). The correlation coefficients $R^2$ are equal
to 0.43 and 0.39 for concentration and size, respectively. This high dispersion can be explained
by the fact that the droplet concentration and size depend also of *LWP*, temperature and the
position of the measurement volume within the cloud system.
At $\lambda = 550$ nm, *IE* = 0.2 and *NE* = 0.43 were obtained. This is to compare with the study of
Garrett et al. (2004) performed at Barrow, Alaska, where *IE* and *NE* were found to be between
0.13 and 0.19 and between 0.3 and 0.5 respectively, at $\lambda = 550$ nm. Very similar values are so
found in two different regions of the Arctic, which confirms these parameterizations of the first
and second aerosol indirect effect for the arctic region.

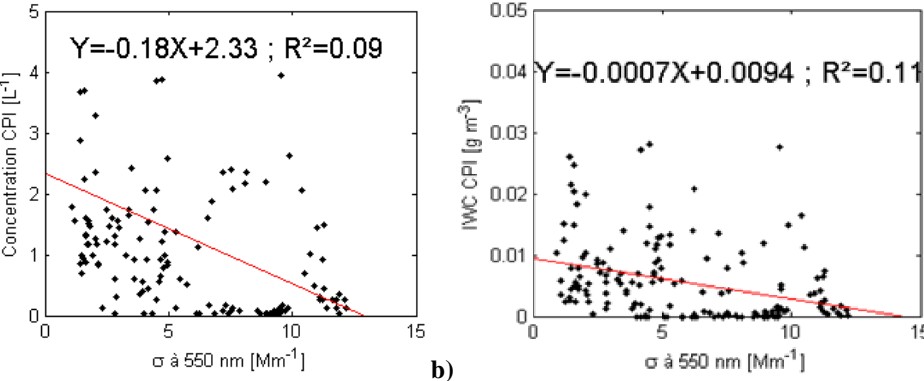

**a)**                                                    **b)**
Figure 14: 5 minutes comparison of the aerosol scattering coefficient $\sigma$ at $\lambda = 550$ nm with a)
the concentration and b) the *IWC* of crystals sampled by the CPI, for the four LMPL cases. A
threshold of 50 µm was applied to the particles diameter to discard the droplets sampled by
the CPI.



The glaciation (Lohmann, 2002a, 2002b) and the riming indirect effect (Borys et al., 2003) were
evaluated during the CLIMSLIP campaign thanks to the CPI and the nephelometer
measurements. The comparison between the crystal concentration and *IWC* with $\sigma$ (or nuclei
concentration) is displayed in Figure 14. The results show very weak correlation, for the
concentration and the *IWC*. This means that neither the glaciation nor the riming indirect effect
were revealed during CLIMSLIP.
This absence can be due to the high uncertainty in the CPI measurements and/or to low sampling
rate that leads to a very low statistical representation. To compare, the study of Jackson et al.
(2012), during the ISDAC and MPACE campaign, found a correlation corresponding to the
glaciation effect above the cloud liquid phase but no evidence of the riming effect.



## *5 Summary and conclusion*

Within the framework of the arctic amplification, the complex interactions between the cloud and aerosol properties remain a challenge to enhance the arctic cloud modeling and to get a better quantification of the consequences of the anthropogenic pollution on the arctic climate. The ANR project CLIMSLIP (CLimate IMpacts of Short-LIved Pollutants in the polar region) provides new data from a ground based aerosol and cloud instrumentation located at the Mount Zeppelin station, Ny-Alesund, Svalbard, during spring 2012. This instrumentation contains a FSSP, a CPI and a Polar Nephelometer to sample clouds and a CPC, a DMPS and a nephelometer for aerosols.

During the campaign, four cases of LMPL (Liquid and Mixed Phase Layer), three cases of snow precipitation layer and two cases of BS (Blowing Snow) were sampled. The precipitation layer cases correspond to the lower layer of a MPC. The precipitation events are composed of large crystals (Mean Diameter $Dm \sim 350\,\mu m$) with an important presence of stellar. The LMPL events are characterized by a bimodal PSD with a large number of droplets. The liquid mode was located around 10 µm and the crystal mode around 250 µm. The phase function measurements showed an increase of the lateral scattering as $F_{liq}$ decreases.

According to Guyot et al. (2015), only isoaxial measurements with a wind speed higher than 5 m/s are selected. This deleted a non-negligible amount of data and so limited the analysis, especially for the precipitation cases where the particle statistics were the weaker. Moreover, the position of the station within the cloud system was approximate despite the ceilometer measurements.

A study by comparison of the effects of the anthropological aerosols transported to the Arctic was performed. According to the FLEXPART/WRF simulations, the "polluted" case of March 29[th] showed air masses from Europe and East Asia whereas the aerosol sources during the "clean" case of March 11[th] were closer (mainly from Scandinavia) and the anthropogenic contribution doesn't exceed northern Europe.

Thus, the polluted case presents higher Black Carbon, aerosol and droplets concentrations, a more important accumulation mode, smaller droplet sizes and higher activation fraction $F_a$. The March 29[th] activation diameter $D_a$ decreased when the droplet diameter increased and $F_a$ increased, proving that smaller aerosol particles are activated and droplets grow up when the aerosol number decreases. These results confirm the first and second aerosol indirect effects with the coefficients *IE* and *NE* respectively around 0.2 and 0.43. These values are very close to those found by Garrett et al. (2004), which performed measurement at Barrow in Alaska, and are so good candidates to be used to parameterize arctic aerosol-cloud interaction in climate models. Furthermore, the crystal concentration and *IWC* do not show any correlation with the aerosol properties, which indicates that the glaciation and riming indirect effects are not highlighted during the CLIMSLIP-NyA campaign.

*Acknowledgements.* This work was supported by the ANR project CLIMSLIP and the conseil general de l'Allier. We also thank the AVI for providing the ceilometer data and the ITM and NILU for monitoring the Mount Zeppelin station. We are grateful to scientists, engineers and technicians that make this study possible. Boris Quennehen acknowledges the IPSL CICLAD/CLIMSERV mesocenter for providing computing resources.



### *Annex: Characterization of the Blowing Snow (BS) cases*
During ground based measurements, some snow was collected that was suspended in the
atmosphere due to wind. This is the so-called Blowing Snow (BS). This annex aims at the
microphysics characterization in order to recognize this kind of episode and the optical
properties measurements, especially the phase function, that can be used as a reference to
develop new parameterizations of the snow simple scattering properties (Räisänen et al., 2015).

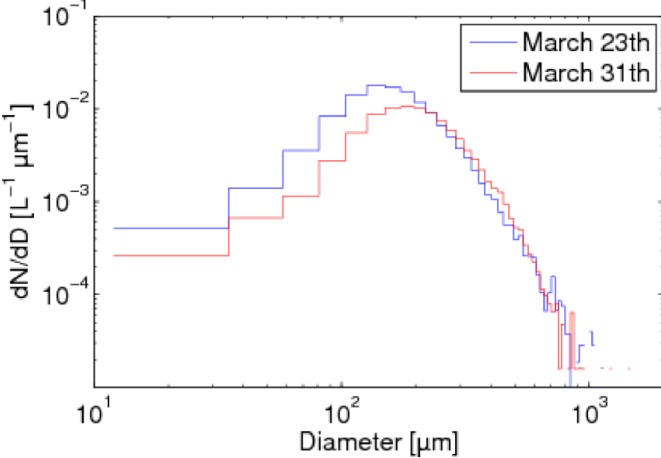

Figure 15: Same as Figure 5 for the BS cases.
When the BS occurs, the sky is clear as observed by the ceilometer. However, crystal particles
are sampled. They are snowflakes initially resting on the ground but getting suspended in the
air by the wind.
Figure 15 shows the average PSDs measured by the CPI for the two BS cases. The shape and
the amplitude are similar for the two PSDs, with a mean diameter between 150 and 200 µm. for
a maximum class concentration around $10^{-2}$ $L^{-1}$ $\mu m^{-1}$. The CPI shape classification, plotted in
Figure 16.a, shows a large prevalence of irregular crystals, as well in number, surface or mass
(i.e. volume), with a percentage around 90% of the crystals. These two characteristics constitute
the microphysics signature of the BS. The difference between the BS and MPC (see Figure 5
and 6) signature makes it possible to identify BS events even if the station is located inside a
cloud.





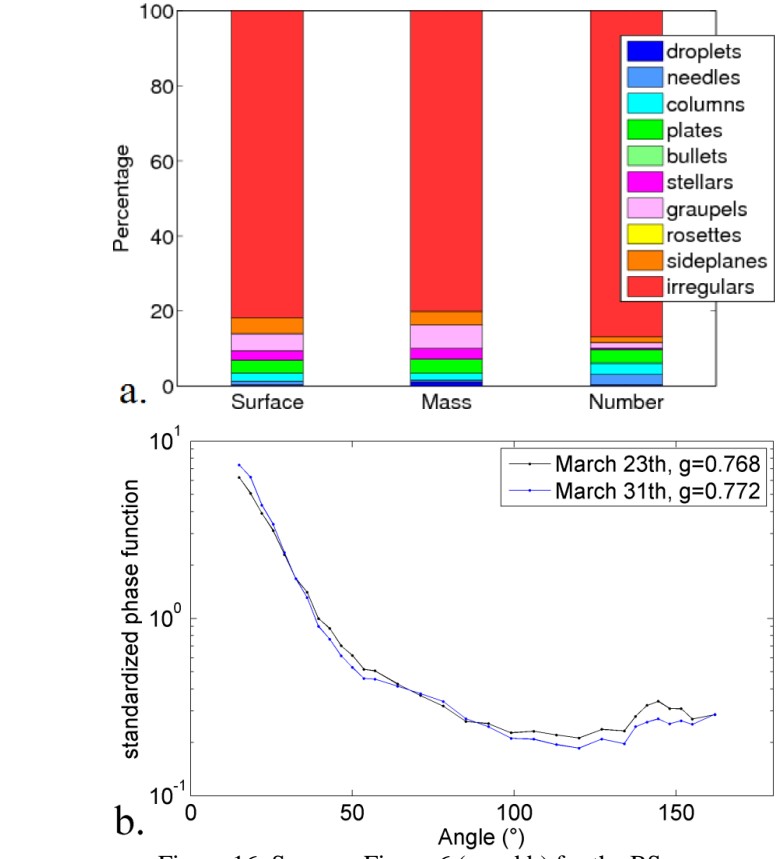


Figure 16: Same as Figure 6 (a and b) for the BS cases.
Even if the resuspension of crystals in the air can modify the shape by impacts, we consider
that the sampled crystals are similar to the deposited precipitations and aged for several days.
Thus, the BS events during CLIMSLIP were excellent occasions to measure the arctic snow
properties.
Figure 16.b displays the average phase functions of the BS cases. The shape of the curves are
very similar to the precipitation cases, typical of ice particles, but with lower $g$ values. These
measurements constitute a unique database to develop parameterizations of the arctic snow
optical properties. Indeed, in most of the climate and weather forecast models, the computation
of the snow albedo uses the approximation of spherical snow grain (Wang and Zeng, 2010).
Thus, the study of Räisänen et al. (2015) proposes new modeling parameterizations of the snow
single scattering properties (SSP) based on the CLIMSLIP-NyA in situ measurement of the
phase function. The obtained snow SSP takes into account the complex BS particles shape
(Räisänen et al., 2015).



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
