# Peer review of "Characterization of the cloud microphysical and optical properties and aerosol-cloud interaction in the Arctic from in situ ground-based measurements during the CLIMSLIP-NyA campaign, Svalbard"

_Atmospheric Chemistry and Physics, 2017_

## Referee Comment (RC1) · Anonymous Referee #1 · 30 Nov 2017

**Review of "Characterization of the cloud microphysical and optical properties and aerosol-cloud interaction in the Arctic from in situ ground-based measurements during the CLIMSLIP-NyA campaign, Svalbard", Guyot et al.**

**Overview of paper:**

This paper presents cloud microphysical and associated measurements from a ground-based site in Svalbard. Three "episodes" are identified, which are conditions where the following was sampled: liquid and mixed-phase layer (LMPL); Precipitation layer; Blowing Snow. These three regimes are characterised in terms of their optical scattering and particle size distribution characteristics.

Next, there is a focus on the cloud characteristics of "clean" vs "polluted" cases. Numerical modelling is used to ascertain source regions; however, from the presentation it is quite difficult to see how these source regions were ascertained. It may be useful to show the backtrajectories.

Lastly  analysis of the indirect effect parameter is presented. Admittedly these show fairly poor correlation, as expected. It is useful, but I think this needs to be presented more clearly because there seems to be a jump in the conclusions when stating that the results confirm the first and second aerosol indirect effects. I did not clearly see how this conclusion comes from the results or discussion.

I am recommending a major revision to this manuscript: the observations are very useful to the community, and the topic is highly relevant; however, the results are not presented in a clear, coherent way and at times I feel the data do not fully support the conclusions / main findings

**General overview / readability comments:**

There are essentially 4 parts to the paper: 1) characterisation of the different episodes that were sampled; 2) numerical modelling to ascertain source region; 3) analysis of clean vs polluted cases; 4) analysis of the indirect effect.

As presented I feel that the modelling does not add a great deal to the main paper, and may be better in supplementary material, which might help give the paper a clearer focus. This may also be the case for the section that characterises the three types of episode. Then the paper could focus on the indirect aerosol effect as its main message. Unfortunately there is a danger that the main message of the paper could be lost because too much is being covered.

I do not have issues with the measurements perse. However, I believe the CPI size distributions are not accurate for particles of sides smaller than 60-100 microns, where there are significant uncertainties. This should be discussed with literature cited to support the discussion.

**Specific comments:**

The title is far too long and unfocussed. I believe the paper would provide the community with a clear message if it focussed on aerosol indirect effects measured from a ground-based site, Svalbard.

There are many typos that will need to be picked up through this manuscript. I have not focussed on picking them up, but it would have to be done before publication.

E.g. Wrong words used in places: introduction line 58 "Specially"….line 76-77: which suspects than clouds… not good gramma – the greenhouse gas effect cannot "suspect" anything

E.g. Contractions used throughout: e.g. "don't " line 117, doesn't line 335 - scientific writing should avoid this.

Abstract and text throughout is often written in the future tense. Past tense is more appropriate for scientific journals

More examples of future tense:, line 239-240: "will be 1 minute for the FSSP…"

Abstract: This is in agreement with the first (Twomey) and second (Albrecht) aerosol indirect effect. I think this statement is inaccurate. I agree that is consistent with droplet activation theory. I am not sure you can argue it is consistent with aerosol indirect effects.

The site is explained along with related measurements. More care is needed here..e.g. a "ceilometer, CL51 model" does not give the manufacturer, Vaisala.  The same is true for the cloud instrumentation. The models are given, but not the manufacturer. Has the PMS FSSP been updated to the latest DMT electronics? Note, the acronym PMS should also be spelled out in full on first usage. In short, more care / attention to the details is necessary here. This is the same throughout these sections. I believe for the aerosol instrumentation section too.

"Due to high discrepancies", line 195. It is not clear what this means to me. High discrepancies in what exactly? And due to what?

Figure 3: should stand alone, but LMPL is not defined in the caption.

Line 282: "the station is so below the mixed layer", not sure "so" is the correct word here

Line 310: talks about true "mixed phase" clouds being rare. This has also be observed from other ground-based sites - See e.g. Lloyd et al.  (2015) , so these papers should be cited.

Section 4.1 the modelling seems to be a small part here. I think it would be better presented in supplementary material, as the only addition they make to the argument is where the air was coming from. However, back trajectories are mentioned, but not presented as far as I could see. The back trajectory plots should be available so that the reader can assess the statements being made.

The explanation of how to find the activation diameter in  section 4.1 could be clearer. "the aerosol PSD is necessary" this is fairly obvious, so why bother complicating the discussion? Just say the DMPS was used to find the diameter where the cumulative number (integrated from right to left) was equal to the drop concentration from the FSSP.

There are two sections labelled 4.1

Section 4.3: this is a key part of the paper. These are the main findings in my opinion. Perhaps these should be the focus of the paper, but yet I still have some misunderstanding.

Your equation 5 says:
$$NE = \frac{\partial \ln y}{\partial \ln x}$$
Your Figure 13 has a curve fit: Y=-0.3X+8.5

If we take the derivative of Y wrt X we obtain:
$$\frac{dy}{dx} = -0.3$$
So
$$\frac{d \ln y}{d \ln x} = m\frac{x}{y} = 1 + \frac{0.3}{y}$$
Therefore by these arguments NE depends on y, and is not a constant. But in the manuscript NE=0.43 is presented. It is not clear to me where the numbers for these coefficients come from. Was a different regression performed that is not in the paper?

---

## Referee Comment (RC2) · Anonymous Referee #2 · 22 Dec 2017

**A review of "Characterization of the cloud microphysical and optical properties and aerosol-cloud interaction in the Arctic from in situ ground-based measurements during the CLIMSLIP-3 NyA campaign, Svalbard" by Guyot et al. (acp-2017-672)**

**General Comments:**

This paper examines cloud microphysical and optical characterization encountered during the ground based CLIMSLIP-NyA campaign performed in Ny-Alesund, Svalbard. Three different scenarios are identified: the Mixed Phase Cloud (MPC), snow precipitation and Blowing Snow (BS) events. Aerosol cloud interaction is also investigated to assess the influence of anthropogenic pollution transported into the Arctic. The results presented in this study are found to be generally consistent with previous studies.

Overall, I found that while the study is reasonably well presented, the English needs to be improved. There are many grammar errors throughout the manuscript (I point out a few in my specific comments below), which warrant a very careful proofread. Also, I have difficulties in finding real new insights in this study. The authors reiterate a few times that their results are (qualitatively) consistent with previous studies, but what is the new contribution that this study makes? This is not very clear to me. In addition, although the study is primarily interested in ground-based in-situ measurements, a description of the synoptic meteorology (perhaps with the help of synoptic charts and some satellite observations) during the events is needed. This will provide necessary information on the synoptic backgrounds under which these events occurred, and will help explain some of the differences observed in the microphysical and optical characteristics. Finally, the uncertainties of some of the analyses need to be better explained.

In my view, a major revision is needed before the manuscript can be considered for publication in ACP.

**Specific Comments:**

**Abstract**

1. Line 33: "presents" should be "present".

2. Line 34: "mostly of" should be "mostly".

**Introduction**

3. Line 56: maybe change "feedbacks" to "feedback".

4. Line 57: "circulation" should be "circulations".

5. Line 65: You say that "The Svalbard region is an exception where MPC are the most frequent cloud independent of season…". But are the clouds over this region representative of the Arctic clouds? I note you comment later on that the aerosols are representative of the aerosol properties in the Arctic (line 93).

6. Line 65 and 69: "MPC" should be "MPCs".

7. Line 75: what do you mean by "umbrella effect"?

8. Line 77: should "suspects" be "suggests"?

9. Line 105: "complexified" should be "complicated".

10. Line 107: "ice content" and "liquid content" should be "ice water content" and "liquid water content".

11. Line 108-109: perhaps replace "cloud cover in lifetime" by "cloud lifetime".

**Site & instrumentation**

This section is good but some clarifications are needed.

12. Line 135: the measurements were made at a Mountain station – are there any orographic effects that could potentially affect the representativeness of the observations? Also, a map showing the location of the stations as well as topography would be useful.

13. Line 179-183: The uncertainties of the classification of particle morphology need to be discussed. You said that a manual classification was also performed but no details are provided. Also, how does this affect your results? No discussion referring to this issue is presented later on.

14. Line 194: "relied related…"???

15. Line 195: "Due to high discrepancies…" high discrepancies of what?

16. Line 229: "…not possible for two reasons not developed further."?? I don't understand this sentence.

17. Line 245: "has" should be "have".

18. Line 246: how does the sampling rate affect the accuracy of the measurements? Please provide some context.

**Identification and characterization of the study cases**

A description of the synoptic meteorology during the events can be presented here (see my general comments). Synoptic charts and some satellite observations (such as MODIS and CloudSat/CALIPSO if available) would be very useful. These satellite observations would provide complementary information that cannot be derived from the ground-based measurements, such as cloud-top temperature/height/phase, etc.

19. Line 278: "On the same time" should be "At the same time".

20. Line 287: "solidification point". Do you mean "freezing point"? Also, you say here that "liquid particles were always supercooled droplet" but previously you say that "No droplets were sampled"? (line 282).

21. Line 305 and Figure 4: it'd be useful to also present the results as a function of temperature to appreciate if the pattern persists across the temperature range. This result can also be compared with previous studies (e.g. Korolev et al. 2003; Ahn et al. 2017).

22. Line 313: suggest insert "for Arctic clouds" after "scientific literature".

23. Line 327 and Figure 5: what causes the large discrepancies between the FSSP and CPI measurements in the overlapping range?

24. Line 334: "in expense" should be "at the expense".

25. Line 334-336: what other mechanisms could potentially explain the ice crystal growth? Are there characteristics of secondary ice production?

26. Line 355: insert "in" before "Figure 6.a.

27. Line 362: "accurately measured"?? But you mentioned before (line 182-183) that a manual classification was also performed due to some malfunctions of the automatic classification?

28. Line 395-399: were these cases sampling different precipitating systems or cloud types (associated with the temperature differences)? Again, synoptic charts and satellite observations could be useful here.

29. Line 399-403: isn't the dynamics (e.g. deep convection vs. shallow convection, convective vs stratiform precipitation) supposed to play an important here? To better understand the temperature effect, as argued by the authors, it is necessary to present Figure 8 as a function of temperature, too. Once again, what are the synoptic processes associated with these events?

**Aerosol-cloud interaction in the Arctic**

30. Line 449: the authors should explain why only the LMPC cases are examined in this section.

31. Line 456-462: did you use a reanalysis dataset to drive the WRF model? What is the vertical resolution of the model? How many levels are within the boundary layer? Quite often the representation of the boundary layer in the model is questionable (partly due to the low resolution). How does this affect the trajectories?

32. Line 619: the assumption of a constant LWP can hardly be valid, especially "the station is in the liquid layer or the mixed phase layer" as you mentioned in line 279-280.

Line 648-653: I'm not sure if the glaciation and riming indirect effect can be evaluated simply using the statistics in Figure 14, as the results can be complicated by scavenging effect, too. Also, how reliable is the IWC measured by the CPI? Further, aerosols may also serve as ice nucleation particles which could facilitate precipitation and ultimately decrease droplet number concentration. This effect is not addressed in the analysis.

**Summary and conclusion**

33. Line 719: insert "for two of the LMPC events" after "was performed".

34. Line 732-733: again, I don't think this conclusion is valid based on the analysis presented.

**Annex: Characterization of the Blowing Snow (BS) cases**

Is there a reason why this analysis can't be a part of the main study?

35. Line 766-767: I don't understand why it is possible. Can't snow develop in clouds or near cloud base?

Ahn E., Huang Y., Chubb TH., Baumgardner D., Isaac P., de Hoog M., Siems ST., Manton MJ. 2017. In situ observations of wintertime low-altitude clouds over the Southern Ocean. Q. J. R. Meteorol. Soc. 143:1381–1394.

Korolev AV, Isaac GA, Cober SG, Strapp JW, Hallett J. 2003. Microphysical characterization of mixed-phase clouds. Q. J. R. Meteorol. Soc. 129: 39–65.

---

## Referee Comment (RC3) · Anonymous Referee #3 · 3 Jan 2018

Goyout and colleagues present an analysis of ground-based observations of thin Arctic mixed-phase clouds, snow precipitating from such clouds, and briefly blowing snow that were observed during the CLIMSLIP4 NyA campaign is Svalbard.

Recommendation:

Continued collection, documentation, and analysis of cloud and aerosol data from Arctic stations are important tasks. The present paper analyzes data from only a small

number of events (4 mixed phase cloud, 3 precipitation, and 2 blowing snow events). Nonetheless measurements in Artic mixed phase clouds are sparse and the paper well worth publication, as it is valuable to have measurements from many sites (to help minimize the possibility of regional biases). While I do have some concerns about the material (comments below), these concerns are reasonably minor. As-is, the paper was (rather obviously) not written by native English speakers. While I was able to follow the text reasonable well (except where noted below) and it is not strictly necessary, I recommend the authors seek additional proof reading.

Overall Recommendation: Minor Revisions.

Note to Editor: While none of my comments are particularly difficult or will require a major revision, as you can see I have a rather long list of comments. I know that some journals require "Minor Revisions" to be completed within two weeks. If that is the case for ACP, you may want to change the recommendation to give the authors more time to respond.

General Comments:

1) Include number of cases and uncertainty.

The analysis is based on only a small number of events. I think you should state the number of cases in the abstract.

Obviously this raises the question of how representative one should consider these results. It would be helpful if the manuscript contained more discussion of how similar (or dissimilar) the results obtained here are to other Artic measurements.

2) Clean vs. Polluted.

I am not sure the "clean" case is really that clean. How do the aerosols in the clean case compare to background conditions (nominally when there is no influence from continental areas)? In general, it might be better to refer to the cases as the "relatively clean" and "relatively polluted" cases.

Specific Comments:

Abstract. Please use present tense when possible.

Line 23. Suggest change to "This study examines cloud microphysical . . .".

Line 28. Strike "these" and change to "In situ cloud measurements are combined with aerosol measurements and air mass back trajectory simulations to study arctic cloud-aerosol interactions."

Line 37. Suggest change to "A relatively polluted case, where aerosol properties are influenced by anthropogenic emission from Europe and East Asia, is compared . . ."

Line 58. Strike "highly".

Line 60. I know this is a minor point, but the analysis of Dufresne and Bony applies to global clouds, and the bulk of the model spread is (I believe) due to changes in tropical and subtropical clouds. The arctic is important, and it would be better to site and discuss studies pointing specifically toward the importance of Artic clouds here (and in particular low-level arctic mixed phase clouds).

Line 69. Suggest change to ". . . are often . . .".

Line 75. I am not sure I understand the intent of the sentence starting "In the Arctic, the umbrella effect . . ." and suggest you simply delete this sentence. Alternatively, you need to expand on your point here, and define what the umbrella effect is."

Line 93. I think "proving" is too strong an ascertain. Perhaps change to "suggesting".

Line 105. Change "complexified" to "complicated".

Line 106. I think you mean "proposed" not "assumed".

Line 129. Strike "if possible". You do quantify several parameters.

Line 145. Perhaps note manufacturer is Vaisala, and provide reference that document instrument performance. Has any research with this instrument be published previously

from this site?

Line 165, Figure 2. "Sampling rate", by which I assume you mean the flow rate, is shown, not sampling volume is given in the figure. Suggest you put sampling volume in figure (and perhaps add short appendix to paper describing calculation based on flow rate if this is not published elsewhere).

Figure 2. I presume units are supposed to be m/s? And the slash or "-1" exponent is missing.

Line 169. Have a fixed alignment seems problematic to me. How often was the alignment changed? Does the data processing include restricting the analysis to periods when the instrument was facing into the wind?

Line 175. I think this uncertainty estimate is somewhat optimistic. Is the presence of (small) ice rather than water an issue for the FSSP measurements? This deserves some discussion.

Line 185. I do not see where the papers by Baker and Lawson conclude the uncertainties on the concentration and the effective diameter are 50 % and 80 %. Please clarify.

Line 192 – 196. I don't think this is a very good description, but in any event, since you don't use the data I suggest you either remove these sentences entirely or simply write "A Nevzorov probe that measures liquid and total water content was also present, but due to concerns with the data quality and inconsistency with the other measurements, these data are not used in this study."

Lines 211-221. Is this the first paper to use this instrument set at the Zepplin Station? This description is rather succinct, and I would generally like to see a more detailed summery with stronger supporting references and discussion of uncertainties, unless this is covered in some other manuscript.

Lines 225. Please elaborate on this. What "conclusions" are you following?

Line 230. How is it that there are more droplets than CCN? Do you mean only during the brief period before noon on March 29? Please explain further.

Line 232. You write, "Thus, this study will not provide quantitative results but qualitative ones based on case comparisons and variation studies." But you DO derive quantities in later section. Suggest you strike this sentence.

Line 234-240. Perhaps move this material up and integrate with earlier discussion of probes. Either that or add some comment to the top if this section indicating you will discuss processing in section 2.2.3.

Limiting CPI data to daily time scale is particularly limiting. What about the time scale for the Neph?

Lines 242-256. Are shattering affects completely avoided? It is not clear to me this is true, how do you know? A variety of correction / shattering detection approaches have been developed (based on the arrival times and looking for bursts of small particles). Are such algorithms being used here? Perhaps better to write "... the advantage that shattering effects may be much smaller."

Lines 254-261. I presume the cases are identified by manual interpretation of lidar imagery and FSSP data, meaning precipitation cases are defined by no significant FSSP signal but the lidar (or CPI or Neph?) shows particles are present? Please clarify.

Also, what was happening at other times? Surely it wasn't entirely "clear sky" conditions, except during these periods. Were there other events when the station was briefly in cloud (but not for long enough to get a good CPI distribution) or events with precipitation falling from higher clouds or deeper systems. Please describe the criteria that led you to these cases.

Line 274. The quoted term "precipitation layer" used here is called "snow precipitation" or simply "precipitation", at various points prior to this line. Please adopt a consistent

term for the entire manuscript.

Line 278. How do you know there is no higher level cloud precipitating through the liquid layer? Perhaps you could use satellite IR and/or sounding data to help ensure these are simple single-low-level cloud layers?

Line 287. What is the "solidification point"? Do you mean you mean the homogenous nucleation temperature for ice of -40 C?

Lines 290-294. Obviously one can simply use the FSSP data to discriminate when you are in a "cloud" (with a small particle mode) or not. Are you suggesting something more sophisticated than this? See comments lines 254-261. If not, suggest you delete these lines, otherwise please clarify.

Line 299. What do you mean by "succesion of layers"?

** Line 305. What is the averaging interval for the CPI? Earlier you seem to indicate only daily averages would be used ... but clearly you have many samples in Figure 4. Have you mixed time scales?

Line 311. I suggest rephrasing this sentence to read, "When liquid water droplets are present, they tend to contribute far more to the total condensate than the ice crystals."

Lines 311- 314. In general, I think it would be more useful to quantify the fraction of time (when liquid is present). That is, how often is Fliq > than 0.9 or 0.95. It seems to me the number of bins you have chosen is somewhat arbitrary and if I add up all the occurrences when Fliq < 0.9 or 0.95 it might be an appreciable fraction of the time?

Line 318. I think I understand what you mean here, but would rephrase as, "One expects a greater range in g for ice particles due to their larger variation in size, as well as shape/habit differences. Thus it is not surprising to observe larger deviations in g (from a pure linear fit) at values of Fliq below 0.5." In general, you might add bars show the standard deviation (or standard error) in g for each category to make it clear this is not simply sampling variability.

Line 325, Figure 4. So the wind speed is always 13.33 m/s? Perhaps this is the mean horizontal sampling interval? I suggest also listing the standard deviation, or some measure of the variability.

** Line 331-336. Costa 2014 is a weak reference (a conference poster) and the link that is provided was not even functioning when I tried to look at the poster. In general, I do not understand what you are trying to communicate. Please expand this point to make the point clear without relying on this weak reference.

Line 340. What do you mean by absolute values? I don't follow. As far as I can see, you are making quantitative use of the IWC and modal values.

Line 355, Figure 6a. Automated classification techniques vary considerably in their classes and results. It would be helpful to see some representative images (good and bad) for the various classes that are heavily occupied. And more broadly, I think researchers might find it interesting to see –and it would be good to document - crystal images from your site. Perhaps you could add such as an appendix?

Line 360. I think you mean "confounded" where you have written "confronted". Please rephrase this sentence.

Line 361. Strike "so".

Line 363. Change to " . . . and a high percentage of regular shapes . . ." .

Line 366. So what should I conclude here? Would you say your results suggest that vapor deposition is NOT the primarily growth mechanism, and its dominated by riming? Or are both important?

** Line 355 - 366. Is aggregation of ice particles important? Where do aggregates fit in the classification? In general one might expect ice effective radius below cloud to increase in size because of aggregation. Is there evidence for aggregation in either you "MPC" or your "precipitating" cases? Why are the particle sizes larger in the precipitating cases.

Lines 376-381, Figure 6b. Why have you focused on the case averages? Given the earlier discussion (Figure 4) regarding the asymmetry parameter and Fliq, I was expecting some analysis of the phase function as a function of Fliq.

Line 383. Suggest you change "lateral diffusion" to "lateral scattering", which you used in the preceeding sentence. Is some distinction here that I am missing? If so, please explain further. Perhaps change to "Indeed, lateral scattering increases when Fliq decreases, as one would expect if there is relatively more scattering by ice crystals and less by water droplets."

Line 408 and 515. Plaque? Perhaps you meant sideplanes??

Line 438-440. See comment lines 290-294. I am not clear on what this procedure is. Please unify this discussion to a simple point in the manuscript (perhaps the concluding section).

Line 438. I think you mean "explored" rather than "explicated".

Line 440-442. I think leaving off a comparison of results from other Arctic regions to a future publication is rather weak (see general comments). It seems to me that such a literature examination can be accomplished in a week, certainly less than two weeks. At a minimum, some discussion of effective radius observed at ISDAC and MPACE should be included.

Line 465. Change "output" to "outputs".

Line 499. Presumable it is only the cloud top you don't know from the lidar. Why do you write that you can't assess the location of cloud base, as well?

Line 513. You have written "DMPS" here, but I think you mean "CPI". Otherwise, something is very wrong with this result.

Line 535. Change to "Northern Europe".

Line 547. Include "possibly". Possibly this is due to scavenging, but might it be due

simply to spatial variability in the aerosol "plume"?

Line 552. Given the description of Da you give on line 554-6, I suggest you show a plot of Da. If not, I suggest you provide some characterization of the variability and reason to believe this change is meaningful.

Line 555. I think "proves" is overly assertive. Rather I would say that it is consistent with expectations of the Towmey effect.

Line 568. Change to "Northern Europe" or otherwise make consistent with early description.

Line 569. Do you mean the aerosol were airborne for a longer period of time = "longer residence time"? Or that they can from farther away? or both? I assume both. Perhaps provide some values for the residence time. Previously you noted the aerosol where "slow moving" in the clean case.

Line 548. Being is not the correct verb tense. Suggest you change to present tense, active voice : "Indeed, McFarquhar et al. (2011), indicate that CCN abilities are mainly due to the aerosol size in the Arctic."

Line 618. I am not sure that assuming the LWP variations are small is a good approximation – and least I don't find your reason compelling. It might help if you provide an estimate for the LWP range for each case, given that you can estimate cloud-base from the lidar, and cloud-top from the radiosonde (I think it is reasonable to assume cloud top at the inversion), and LWC from the FSSP. Obviously with only 4 cases you can't afford to restrict the analysis. But I don't think you should just dismiss this factor. Rather (unless the above estimated do fall in a narrow range) I think you are simply going to state it is a limitation of your analysis and provide estimates for the LWP.

\*\*Line 635. The values for IE and NE shown here are NOT the slopes shown in Figure 13. Shouldn't these data be on a log-log plot?

Lines 637-639. Perhaps change to, "These values for IE and NE are similar to those

and reported by Garrett et al. (2004) and support his contention that these values are higher in the arctic region than at lower lattitudes."

Line 709. Change "stellar" to "stellars".

Line 713. Change "According" to "Following".

Line 757. If I understand, the lidar is not at the site but in the nearby town. Is it possible there was cloud over the mountain and not over the town? Are there notably higher wind speeds for these cases ?

Line 765. To me, the important question is not whether you can tell the difference between being cloud and blowing snow, but can you tell the difference between "precipitating snow" and "blowing snow". Is the only difference that blowing snow is composed of a large fraction (> 90%) irregulars? That is helpful, but doesn't this leave open the possibility that precipitating events composed of heavily rimed particles will look just like blowing snow by this criteria.

Line 773. I don't understand the intent of this sentence regarding resuspended particles. Please rephrase to make the relevance clearer.